# Transgenerational Effects of Water-Deficit and Heat Stress on Germination and Seedling Vigour—New Insights from Durum Wheat microRNAs

**DOI:** 10.3390/plants9020189

**Published:** 2020-02-04

**Authors:** Haipei Liu, Amanda J. Able, Jason A. Able

**Affiliations:** School of Agriculture, Food & Wine, Waite Research Institute, The University of Adelaide, Urrbrae SA5064, Australia; amanda.able@adelaide.edu.au (A.J.A.); jason.able@adelaide.edu.au (J.A.A.)

**Keywords:** abiotic stress, transgenerational effect, microRNA, seedling vigour, durum wheat, crop improvement

## Abstract

Water deficiency and heat stress can severely limit crop production and quality. Stress imposed on the parents during reproduction could have transgenerational effects on their progeny. Seeds with different origins can vary significantly in their germination and early growth. Here, we investigated how water-deficit and heat stress on parental durum wheat plants affected seedling establishment of the subsequent generation. One stress-tolerant and one stress-sensitive Australian durum genotype were used. Seeds were collected from parents with or without exposure to stress during reproduction. Generally, stress on the previous generation negatively affected seed germination and seedling vigour, but to a lesser extent in the tolerant variety. Small RNA sequencing utilising the new durum genome assembly revealed significant differences in microRNA (miRNA) expression in the two genotypes. A bioinformatics approach was used to identify multiple miRNA targets which have critical molecular functions in stress adaptation and plant development and could therefore contribute to the phenotypic differences observed. Our data provide the first confirmation of the transgenerational effects of reproductive-stage stress on germination and seedling establishment in durum wheat. New insights gained on the epigenetic level indicate that durum miRNAs could be key factors in optimising seed vigour for breeding superior germplasm and/or varieties.

## 1. Introduction

Global food demand is constantly increasing with the rise in our world population. However, even with a finite amount of arable land, agricultural production is still expected to provide more food and fibre, as well as to adapt to the environmental challenges (i.e., climate resilience). High quality seeds or grains are therefore essential for sustainable agriculture and food security. Cereal crop production starts with seed germination to establish vigorous healthy seedlings in the field. Successful germination and robust seedling growth are critical early stages of crop life to ensure reliable canopy establishment and production efficiency [1,2]. Seeds with high vigour are expected to germinate rapidly and uniformly, thus warranting field emergence with minimum delay after sowing [3]. Seedlings with high vigour are expected to be more resource-use-efficient and to have competitive advantages over unwanted weed species [1,2].

For cereal crops, successful germination and vigorous seedling emergence relies on high quality seeds. Durum wheat (*Triticum turgidum* L. ssp. *durum*) is a vital cereal crop that provides substantial economic output in the Mediterranean region [4] due to its unique use in multiple food products such as pasta and couscous. Durum is better adapted than common hexaploid wheat to semi-arid environments, but durum wheat production and quality suffer greatly from several abiotic stress constraints including water deficiency and heat [5,6]. Many studies have investigated the impact of water-deficit stress and heat stress on yield and quality, with emphasis on grain traits, such as protein content and carbohydrate composition, that are directly associated with end use (e.g., pasta-making) [7,8,9]. However, there is no evidence of research to date that has focused on investigating the transgenerational effects of stress on seed vigour and seedling establishment in durum wheat. Previous studies in other crops with agronomic significance have shown that the effects of environmental stresses can be transmitted to the offspring with significant impact [10,11,12,13]. A recent study in soybean reported that exposing parental plants to drought stress significantly reduces germination and seedling growth of the F1 generation grown under optimum conditions [11]. In rice, transient heat stress shortly after fertilisation can either promote or supress seed germination rate, depending on the timing of the stress [12]. In canola, drought stress on the mother plants actually increased the progeny seedling vigour, independent of the maternal yield performance [10]. Thus, establishing the transgenerational effects of abiotic stress in durum wheat is of relevance, particular for the broader wheat breeding programs that exist globally, where such information could be translated to enhance breeding strategies of the future.

Seeds originating from different parents not only inherit their full genetic components, but also their epigenetic marks [14]. Rapidly accumulating evidence shows that epigenetic modification can have transgenerational effects which have an impact on adaptive traits in plants [15,16]. Small RNAs (sRNAs) are a class of non-coding RNAs with essential regulatory roles in the epigenetic phenomena [17]. In plants, the major sRNA types include microRNAs (miRNAs) and small-interfering RNAs (siRNAs) [18]. Plant miRNAs can precisely reprogram the expression of their protein-coding target genes by inducing mRNA degradation or translational inhibition through post-transcriptional gene silencing [17,18]. In crops, miRNAs can rapidly respond to environmental stress and act as mobile signal molecules to help restore cellular homeostasis [19]. Numerous studies have shown that miRNAs are important for crop plants in growth, development and stress adaptation [20,21,22,23,24,25,26,27,28]. The involvement of miRNAs in stress memory has also been demonstrated as part of the within- or cross-generation phenotypic plasticity [16,29]. Next generation sequencing (NGS) or microarray technologies have been used to study the miRNA profile in durum wheat under various types of abiotic stress [20,21,22,30,31,32]. However, most efforts have focused on the same generation of stress occurrence. We lack the knowledge of the transgenerational impact of stress on the miRNA transcriptome (miRNAome) in progeny, and its possible implication associated with early seedling establishment. 

In this study, we therefore aimed to elucidate the transgenerational impact of water-deficit and heat stress on germination and seedling vigour of stress-tolerant and stress-sensitive durum genotypes. To understand the molecular mechanism underlying altered genotypic performance, sRNA sequencing was used to profile the miRNAome along with *in silico* characterisation of their functional target genes. To our knowledge, this is the first description of the transgenerational effects of abiotic stress in durum wheat integrated with genome-wide sRNA sequencing. Moreover, the current study is the first to employ the new durum genome assembly (Svevo.v1, available 2019 [33]) for epigenetic studies, providing a more accurate, in-depth analysis on the whole-genome scale. 

## 2. Results 

### 2.1. Evaluation of Germination and Seedling Establishment

Two Australian durum wheat genotypes were used in this study in controlled environment facilities. DBA Aurora is a leading Australian commercial variety that has shown high tolerance to pre-anthesis water-deficit stress and post-anthesis heat stress [8]. Line 6 (L6) is an advanced University of Adelaide breeding line (UAD1301020-8) that is sensitive to the combination of water-deficit and heat stress [8]. The seeds used in this study were collected from the parents that were in the control group and the water-deficit and heat stress group of both genotypes [8]. In total, there were four different seed groups: AuCG (seeds from DBA Aurora control group), AuWH (seeds from DBA Aurora water-deficit and heat stress group), L6CG (seeds from L6 control group), and L6WH (seeds from L6 water-deficit and heat stress group). 

For seed germination measurements (Table 1), genotype had a significant impact (*p* < 0.05) on the germination potential (Gp), germination rate (Gr), and germination index (Gi) without interaction with the parent treatment factor. Overall, DBA Aurora had better germination performance than L6. In particular, AuCG and AuWH almost had double the germination potential of L6CG and L6WH. Parent treatment only had a significant impact (*p* < 0.05) on Gp and Gi without interaction with the genotype. Overall, water-deficit and heat stress treatment on the parents negatively affected the germination performance of the next generation. Coleoptile length was not significantly affected by any of the factors. Interestingly, a significant interaction (*p* = 0.005) was detected between genotype and parent treatment for young root length. For DBA Aurora, the parent stress treatment did not significantly affect the young root of the progenies. However, for L6, stress on the parents significantly reduced the length of the young roots. 

For the seedling vigour index I (SVI-I) and seedling vigour index II (SVI-II) (Table 2), both genotype and parent treatment had a significant impact (*p* < 0.05) without interaction with each other. Overall DBA Aurora outperformed L6. Water-deficit and heat stress (WH) treatment of the parents negatively impacted both genotypes, but to a greater extent in L6. For other seedling measurements (Table 2), genotype had a significant impact (*p* < 0.05) on shoot fresh weight, shoot dry weight, and seedling height, with no interaction with the parent treatment factor. Overall, DBA Aurora again displayed a better performance for these traits when compared to L6. Parent treatment also had a significant impact (*p* < 0.05) on shoot fresh weight and seedling height without interaction with the genotype factor. The WH treatment of the parents significantly reduced the shoot fresh weight and seedling height of the next generation (*p* < 0.05). Interestingly, a significant interaction (*p* < 0.05) was detected between genotype and parent treatment for all root traits (root fresh weight, root dry weight and root length). For DBA Aurora, the WH treatment of the parents had no impact on the root fresh weight and dry weight of the next generation seedlings. For L6, the WH treatment of the parents significantly reduced the root fresh weight and dry weight. The WH treatment of the parents promoted the root length in both genotypes, but to a greater extent in DBA Aurora.

### 2.2. The microRNAome in the Progenies

Four sRNA libraries (TCG, TWH, SCG and SWH) were constructed and sequenced from the four seedling groups (AuCG, AuWH, L6CG and L6WH), respectively. A total of 83,991,830 raw reads were obtained (Table 3). Approximately 52.04 million clean sRNA reads within the 18–25 nt range remained after data processing. Among these, 26.16 million non-redundant unique sRNA reads were obtained. The size distribution of sRNA reads exhibited a two-peak pattern (a weak peak at 21 nt and a strong peak at 24 nt) (Figure 1), a classic profile expected for cleaved products of plant Dicer-like enzymes [20,22,34]. 

A total of 2202 MIR-miRNA entries (different combinations of MIR loci origin and mature miRNA products) were identified (Appendix A). All MIR-miRNA entries were categorised into five groups (G1–G5), with G1–G4 representing conserved miRNAs and G5 representing novel miRNAs (Table 4). The sequence reads that mapped to the durum genome and ESTs (expressed sequence tags) but their extended genome sequences could not form secondary hairpins (G3) had the highest number of unique miRNAs. The group that had neither aligned reference pre-miRNAs nor sequencing reads that could be further mapped to the genome (G4) had the lowest number of unique miRNAs (Table 4). In Appendix A, MIR-miRNA entries with the same pre-miRNA index number within each group share the same pre-miRNA sequence. As one pre-miRNA can potentially produce multiple unique mature miRNAs, each unique miRNA is labelled with its own miRNA index number within each group. The pre-miRNA and miRNA index numbers in Appendix A can be linked to Table 4. For Group 2 to Group 4, the total number of pre-miRNAs and the total number of unique mature miRNAs correspond to the highest pre-miRNA index number and the highest miRNA index number within each group in Appendix A (e.g., the last MIR-miRNA entry for Group 2 in Appendix A is indexed as 446–483). However, the highest miRNA index number in Group 1 (Appendix A) is different to the total number of unique miRNAs in Group 1 (Table 4) as the same mature miRNA (unique miRNA) can be produced from different pre-miRNAs with different index numbers.

Among all entries (Appendix A), 200 miRNAs were highly expressed (normalised reads number of the reported miRNA is higher than the average of the dataset), including 190 conserved miRNAs and 10 novel miRNAs. A total of 1435 miRNAs showed a medium expression level (normalised reads number of the reported miRNA is higher than 10 but less than the average of the dataset), including 1002 conserved miRNAs and 433 novel miRNAs. A total of 567 miRNA entries were lowly expressed (normalised reads number of the reported miRNA is less than 10), including 525 conserved miRNAs 42 novel miRNAs. In total, these 2202 MIR-miRNA entries represent 2177 unique mature miRNA products among four libraries, as shown in the Venn diagram (Figure 2). Collectively, the conserved miRNAs identified in this study represent 70 MIR families (Appendix A). The conservation profiles of all miRNAs across the reference plant species (Appendix A) in miRBase are also shown (Figure 3).

### 2.3. Differentially Expressed miRNAs (DEMs)

Differentially expressed miRNAs (DEMs) were identified based on their normalised reads number. A total of 1368 miRNAs (Appendix A) showed differential expression across four libraries (TCG vs TWH vs SCG vs SWH). A heatmap was compiled to show DEMs across four libraries using their log10 (normalised count) value (Appendix A). Some miRNAs exhibited a genotype-specific pattern (Appendix A). A total of 144 miRNA were found to be exclusively expressed in DBA Aurora (TCG and/or TWH), while 139 miRNAs were only expressed in L6 (SCG and/or SWH). A number of miRNAs also showed specificity to parent treatment (Appendix A). A total of 129 miRNAs were only detected in the CG libraries (TCG and/or SCG) while 72 miRNAs were only found in the WH libraries (TWH and/or SWH).

When comparisons were made between two libraries within the genotype or within the treatment source (TWH vs TCG, SWH vs SCG, SCG vs TCG, SWH vs TWH), DEMs were identified using Chi-squared 2 × 2 test and Fisher’s test (see Materials and Methods). The number of up- and down-regulated DEMs identified in each comparison using different *p* values as thresholds are shown in Figure 4. Interestingly, a genotypic pattern can be observed here for the number of DEMs. For the stress-tolerant genotype DBA Aurora (TWH vs TCG), the number of miRNAs that were up-regulated was higher than the ones that were down-regulated, irrespective of the *p* value used. However, for the stress-sensitive genotype L6 (SWH vs SCG), an opposite trend was observed, where there were more miRNAs that were down-regulated than up-regulated. When the same parent treatment source was compared between two genotypes (SCG vs TCG, SWH vs TWH), there are more down-regulated miRNAs than up-regulated miRNAs (except for SCG vs TCG at *p* < 0.1). The expression levels of all DEMs are shown in Appendix A. Highly expressed DEMs with a |log2 (fold change)| > 1 in each comparison (*p* < 0.05) are listed in Table 5.

### 2.4. Target Prediction and Enrichment Analysis 

The target genes were predicted for highly expressed DEMs using *in silico* analysis with the recently constructed durum wheat genome (see Materials and Methods). A total of 311,964 transcripts (target genes) were identified as potential targets for 189 DEMs (Appendix A). Gene ontology (GO) analysis was conducted to classify the biochemical functions of the predicted target genes. GO annotation grouped all targets in the following three GO categories (i) Biological Process (BP), (ii) Molecular Function (MF), and (iii) Cellular Component (CC). GO enrichment analysis of the predicted targets was performed (Appendix A). The most represented GO terms are shown in Figure 5. Notably, a few biological processes with importance are highly represented, including the regulation of transcription, DNA-templating (GO:0006355), protein phosphorylation (GO:0006468), the oxidation-reduction process (GO:0055114), transmembrane transport (GO:0055085) and signal transduction (GO:0007165). Highly represented cellular components include the plasma membrane (GO:0005886), cytoplasm (GO:0005737), integral components of the membrane (GO:0016021), cytosol (GO:0005829) and chloroplast (GO:0009507). In the molecular function category, binding activities are highly represented, including protein binding (GO:0005515), ATP binding (GO:0005524), DNA binding (GO:0003677) and metal ion binding (GO:0046872).

KEGG (Kyoto Encyclopedia of Genes and Genomes) pathway enrichment analysis was performed as previously described [35] to identify the biochemical pathways with which durum target genes were associated (Appendix A). The top-ranked enriched pathways are shown in Figure 6. In particular, a total of 2406 targets were matched to protein processing in endoplasmic reticulum (ko04141), 2248 targets were matched to starch and sucrose metabolism (ko00500), 2171 targets were matched to the MAPK signalling pathway (ko04010), 1191 target were matched to the spliceosome (ko03040), and 1720 targets were matched to ABC transporters (Appendix A).

## 3. Discussion

Cereal crops are often challenged by multiple biotic and abiotic factors during their life cycle, affecting both production volume and quality [8,36,37,38,39]. The detrimental effects on the crop plants are often exacerbated when several stress factors are combined, simultaneously or sequentially [8,40]. Increasing evidence has shown that the effects of environmental stress on the parental plants can be carried over to the subsequent generation(s). In the attempt to achieve sustainable crop production, it is essential that we understand the impact of abiotic stress within and across generations, as well as the fundamental networks that underpin crop performance. The influence of the maternal environment on the offspring early growth has rarely been explored in durum wheat before the current study.

For the two genotypes used in this study (DBA Aurora and L6), the parental exposure of water-deficit and heat stress during the reproductive stage significantly reduced the germination potential and germination index. These traits reflect the time-course and the cumulative status of seed germination. Similarly, in soybean, water shortage during the reproductive stage had a pronounced transgenerational effect on seed germination performance [11]. The cumulative percentage of germination negatively correlated with the severity of the drought stress that was imposed on the parents. Germination rate was also significantly reduced in the F1 generation, with reduced seed number and individual seed weight in both soybean cultivars. However, in our study, the germination rate was not affected by the parental stress treatment factor. Moreover, the genotype had a significant impact on all germination traits measured (except for the coleoptile length), with superior performance observed for DBA Aurora. In our previous study, the effect of stress on the parental plants, including physiological responses, yield component and particularly grain traits, was well documented [8]. DBA Aurora produced bigger grains that were full-looking under water-deficit stress and heat stress. The 1000-grain weight of DBA Aurora water-deficit plus heat stress group was significantly higher than that of the control group. On the other hand, L6 had significantly smaller seeds with reduced weight under the effect of such stress (that is, significantly lower 1000-grain weight compared with its control group). For cereal species, high temperature and water shortage during reproduction often leads to an accelerated grain growth rate and a shortened grain development period [40]. As a result, smaller grains with poor grain filling are often produced. However, elite stress-tolerant varieties like DBA Aurora could mitigate the adverse impact of stress and still manage to produce bigger grains, which could (at least in part) contribute to its superior germination performance over L6. It is also worth noting that there is no significant difference between the 1000-grain weight of L6WH seeds and AuWH seeds, suggesting that other factors could contribute to the transgenerational effect of stress rather than grain size.

Similar to germination traits, early durum seedling growth of the offspring was also significantly affected by the parental stress exposure. Genotype seems to play a significant role here. Although seedling vigour was significantly reduced under the transgenerational effect of stress in both genotypes, a greater impact was always observed in L6. In particular, a significant interaction was found between genotype and parent treatment for all root traits measured in the young seedlings. Only L6 has shown reduced root weight due to parental stress exposure. Such genotypic differences in the transgenerational effects of stress have also been noted before [41]. The findings suggest that breeders and growers should take care in selecting varieties under different parental environments for crossing or seed production (this is often overlooked, with a ‘mother stock’ source used for such purposes). Furthermore, the stress exposure of the parents increased the young seedling root length for both genotypes (with a higher increase in DBA Aurora), suggesting that root elongation could be promoted through transgenerational stress exposure irrespective of genotypes. Establishing a longer, deeper root system at earlier developmental stages might be an adaptive mechanism to prepare the offspring to improve water and nutrient uptake in any possible future stress occurrence [11]. Previous research has suggested that the impact of parental stress on early seedling growth is often associated with seed storage reserves such as carbohydrates and proteins. However, in our case, both the DBA Aurora and the L6 parents produced grains with increased grain protein content and reduced starch content under the same stress without any significant genotypic difference [8]. In the miRNA-target analysis, starch and sucrose metabolism came second for the most-enriched KEGG pathways (Appendix A). The 1639 targets enriched for this pathway included many enzymes, such as sucrose synthase, starch synthase, beta-glucosidase, sucrose phosphatase, and isoamylase (Appendix A). Altered enzyme activities regulated by differentially expressed miRNAs can affect how storage reserves were utilised during germination and early growth. The results suggest that, apart from seed quality traits being affected by stressful paternal environments, epigenetic mechanisms such as miRNA-target modules could be in play during the transmission across generations [11]. 

As a central class of epigenetic regulators, miRNAs take part in a wide range of cellular biological processes [17,18,42]. Hence, the identification and annotation of the miRNAome enables the epigenetic networks underlying the transgenerational effect of stress to be illustrated. The combination of NGS and bioinformatics analysis using genome reference has been the most effective approach to characterise the miRNA population on a genome-wide scale [17]. Such technologies have led to the complete discovery of small RNAs in many plant species on an unprecedented scale. However, despite a few reports in durum wheat (including our previous studies) [20,22,30,34], the discovery of novel miRNAs and a fully comprehensive miRNAome description in this important cereal has not been previously achieved due to incomplete genome information. Only recently, the ground-breaking genome assembly of the modern durum cultivar Svevo was made available [33]. The complete *Triticum turgidum* genome presented us with a timely opportunity to harness additional information from sRNA sequencing datasets. For the first time, to our knowledge, a systematic and complete archive of durum miRNAs has now been provided. In total, we successfully identified 1717 conserved and 485 novel miRNA entries matched with their MIR loci origins. This number exceeded all previous reports in durum wheat. Interestingly, among all the conserved miRNA entries, the top plant species with the highest miRBase conservation match was *Glycine max*, followed by *Oryza sativa*. Hexaploid wheat (*Triticum aestivum*), with tetraploid wheat as one of its genome progenitors, only came third in conservation ranking. This is likely due to soybean miRNAs being very well-documented (684 precursors, 756 mature) in the miRBase when compared to bread wheat (122 precursors, 125 mature), which is not surprising considering that the *Glycine max* genome was well-annotated ahead of all wheat species sequenced. Notably, there is currently only one *Triticum turgidum* miRNA entry (ttu-miR160, MI0016454) deposited in the miRBase [43]. It could be foreseen that this number will increase rapidly with the first durum genome assembly now being available to the scientific community.

Plant miRNAs can act rapidly in response to both environmental and developmental signals [17]. In particular, various studies have described the functional roles of miRNA-target modules in combatting water deficiency and extreme temperatures in cereal crops [8,23,24]. In our previous research, we have demonstrated that the expression of durum miRNAs can change differently with different timings of stress adaption within the life cycle [23,24]. Such patterns were also genotype-dependent among stress-tolerant and -sensitive varieties, suggesting that miRNA-mediated regulatory modules could contribute to the genotypic differences in the physiological performance, ultimately affecting yield production [23]. The present study further illustrated that durum miRNAs behave in a genotypic manner under the effect of transgenerational stress. Overall, there were more up-regulated miRNAs (275, *p* < 0.05) than down-regulated miRNAs (181, *p* < 0.05) in the stress-tolerant genotype, but more down-regulated miRNAs (242, *p* < 0.05) than up-regulated (153, *p* < 0.05) miRNAs in the stress-sensitive genotype. Durum miRNAs that were induced by the transgenerational effects could repress the expression of protein-coding genes with negative regulatory roles, while reduced miRNA expression could promote the expression of target genes to increase protein products with positive functions. A higher number of induced miRNAs in the stress-tolerant genotype and higher number of repressed miRNAs in the stress-sensitive genotype suggest that contrasting mechanisms were preferably deployed in response to transgenerational stress, resulting in the difference in young seedling growth.

Investigations into the expression pattern of specific durum miRNAs, together with the molecular functions of their predicted targets, provide further information on the possible pathways activated in different genotypes. As an example, tae-miR1847-5p was significantly down-regulated in DBA Aurora (−0.90 fold) under the transgenerational effects of stress (Appendix A). *In silico* analysis revealed that tae-miR1847 targets several CREB-binding protein-like family members located on the 6A and 6B chromosomes (Appendix A). CREB-binding proteins (CBPs) are a type of transcriptional coactivator that can switch on the cAMP-dependent signalling pathways, which are key components employed by many hormone-regulated activities to modulate gene expression [44,45]. CBPs initiate the function of many classes of transcription factors through their intrinsic transferase activity to acetylate histones, which is a key process associated with chromatin remodelling [44]. Stress-related chromatin modification plays a critical role in transgenerational memory in plants [15]. The repressed expression of tae-miR1847 in the stress-tolerant genotype would increase the expression of its target CBPs, which could ultimately promote histone acetyltransferase-regulated transduction pathways. However, this pathway might not be in play in the stress-sensitive genotype, as tae-miR1847-tp was up-regulated (0.23-fold) under the transgenerational effects of stress. Another example of a miRNA-regulated module that could contribute to the differential seedling performance involved mitogen-activated protein kinase (MAPK). MAPK signalling is one of the top KEGG pathways enriched among the predicted targets. MAPK cascades are a universal signal transduction mechanism that play diverse roles in connecting intra- and extra-cellular crosstalk in eukaryotes [46]. MAPKs are involved in virtually all the important aspects of plant development and response to environmental signals. In particular, MAPKs facilitate plant adaptation and survival under many abiotic stresses, including drought, heat and salt stress, via coordinating the transduction of secondary messengers and hormone signals [46,47]. Here, a few durum miRNAs that target MAPKs exhibited genotype-dependent expression patterns. For example, conserved miRNAs such as ttu-miR160, hvu-miR444b, and tae-miR1847-5p were all down-regulated (−0.42-fold, −0.45-fold and −0.90-fold, respectively) in DBA Aurora under the effect of transgenerational stress (Appendix A). This promotes the expression of MAPKs in the stress-tolerant variety to facilitate signal transduction. However, the expression of these miRNAs was either unchanged (ttu-miR160 and hvu-miR444b) or increased (tae-miR1847-5p) in L6, suggesting that miRNA-MAPK modules were not activated in the stress-sensitive genotype. Research in bread wheat has also demonstrated the involvement of MAPK in transgenerational tolerance to heat stress [48]. Increased expression of MAPK in the progenies of heat-stressed parents contributed to the enhanced activation of Ca^2+^ signalling pathways to facilitate stress adaptation [48]. Moreover, stress-induced MAPK could promote root tip growth through controlling the organisation of the actin cytoskeleton [49]. Durum miRNA-promoted MAPK expression could possibly contribute to the adaptive root traits observed in the stress-tolerant genotype. Future research should investigate the co-expression of durum miRNAs and MAPK family members in different tissues types to further our understanding of miRNA-MAPK modules in transgenerational stress adaption. 

The spliceosome (ko03040) is another major KEGG pathway enriched for the targets of differentially expressed durum miRNAs. The pre-mRNA splicing machinery is governed by the spliceosome, a large ribonucleoprotein complex that recognises the pre-mRNA splice sites to remove introns and join exons [50]. Research has shown that alternative splicing (AS), which enables the production of multiple mRNA isoforms from one single gene, is closely associated with abiotic stress adaption [51,52]. The AS process provides a fine-tuning mechanism to regulate the levels of active and inactive isoforms of stress-associated proteins. The specificity of splice-site selection in AS largely relies on serine/arginine-rich (SR) proteins (splicing enhancer) and heterogeneous nuclear ribonucleoproteins (hnRNPs, splicing suppressor) [53]. In our study, a large number of SR and hnRNP family members were targeted by differentially expressed durum miRNAs. Some miRNAs exhibited a genotype-dependent expression, while others seem to have universal regulatory roles. For example, ttu-miR160 targets several hnRNP-encoding genes located on the 4A and 4B chromosomes (Appendix A). It also targets thirteen SR splicing factor-coding genes located on chromosome 6A and 6B. ttu-miR160 was down-regulated in DBA Aurora (−0.42-fold) in response to transgenerational stress, indicating that the expression of SR proteins and hnRNPs were possibly increased to coordinate the AS process. ttu-miR160-regulated AS was not in play in the sensitive genotype, based on the unaltered miRNA expression. However, a few other miRNA-regulated modules were active in both genotypes. For example, tae-miR9772, ata-miR396c-5p, and a miR396c variant (sbi-miR396c_R+1) all target hnRNP 1-like genes. The expression of these miRNAs was down-regulated in both genotypes, suggesting that hnRNP-modulated splicing events were activated by miRNAs without genotypic specificity. Furthermore, stress-induced AS can regulate the processing of non-coding RNA transcripts, so as to control the production of different pre-miRNA hairpins [54]. In barley, heat stress induced the splicing of introns hosting the miRNA primary transcripts (pri-miRNAs) of miR160a and miR5175a, which was correlated with the accumulation of mature miRNA products [55]. The results suggest that the durum miRNA-modulated AS mechanism could possibly provide a feedback loop to fine-tune the post-transcriptional regulation of miRNA precursor processing. Looking forward, the complexities in durum miRNA expression, spliceosomal protein activities, and the production of pri-miRNA isoforms require further exploration. In addition, it would be desirable for future work to study the upstream regulatory mechanisms of durum miRNAs so as to understand the underlying reasons for miRNA expression changes between stress-tolerant and stress-sensitive genotypes. Genomic analysis of the surrounding regions (such as transcription start sites and *cis*-regulatory motifs) of major stress-responsive miRNAs can compare the promoter features of the stress-tolerant and stress-sensitive durum genotypes to provide valuable information on transcriptional regulation of MIR genes. Our current research has paved the way for future investigation, having provided in-depth details of the MIR gene (genome ID, length, pre-miRNA sequence, secondary structure etc., Appendix A).

In conclusion, the present study provides a valuable resource for an improved understanding of the transgenerational effects of water-deficit and heat stress on germination and seedling growth in durum wheat, as well as new insights of conserved and novel miRNAs that are involved in the epigenetic regulatory networks. Such knowledge provides a new foundation for durum researchers and breeders to use effective strategies for developing and selecting elite varieties that can mitigate or adapt to the negative impacts of stress that are transmitted across generations. 

## 4. Materials and Methods 

### 4.1. Plant Materials

Seeds of two Australian genotypes were used in this study. The seeds of both genotypes were collected from the parental plants in the control group and the water-deficit and heat stress group from the previous year’s experiment [8]. Briefly, water-deficit stress was applied to the parents from the booting stage by maintaining the soil water content at half of the field capacity. Heat stress was applied post-anthesis by exposing the plants to 37/27 °C for 24 h at 5, 15, 25, 35, and 45 days after flowering (a total of five stress episodes on the same plants). In total, four different seed groups were used in this study—AuCG (seeds from DBA Aurora control group), AuWH (seeds from DBA Aurora stress group), L6CG (seeds from L6 control group), and L6WH (seeds from L6 stress group).

### 4.2. Germination Test

Germination tests of the four seed groups were performed according to the International Rules for Seed Testing 2019 (https://www.seedtest.org). The seeds used for germination were randomly selected. The seeds were sterilised by rinsing with 70% ethanol for 5 min, followed by a wash with sterile distilled water. The seeds were germinated on filter paper in petri dishes containing sterile distilled water for 96 h at 22 °C. Three replicates per group were used with 30 seeds per replicate. germination potential (Gp, %) was determined as number of seeds germinated within 24 h/total number of seeds × 100%. The germination rate (Gr, %) was determined as number of seeds germinated within 96 h/total number of seeds × 100%. The germination index (Gi) was determined as ∑(n_t_/d_t_), where n_t_ represents the number of germinated seeds on the t day, and d_t_ represented germination days [56]. The length of coleoptile and young roots was recorded at 96 h. 

### 4.3. Measurement of Seedling Establishment

For each group, 16 randomly selected germinated seeds were transferred into pots (one seedling per pot). Each pot contained 1.2 kg of N40 sand with 0.5% CaCO_3_, and nutrient solution was provided as previously described [8]. The seedlings were grown under controlled glasshouse conditions with day/night temperature of 24/18 °C and a photoperiod of 16/8 h. Seedling height, shoot fresh weight and root fresh weight were measured on 12-day old seedlings. Shoot dry weight and root dry weight were measured after overnight oven drying at 65 °C. Fresh shoots were collected and snap-frozen in liquid nitrogen for RNA extraction. Eight biological replicates were used for seedling measurements and eight biological replicates were used for RNA extraction. Seedling vigour index I (SVI-I) was determined as germination rate × seedling fresh weight in mg [57]. Seedling vigour index II (SVI-II) was determined as germination rate × seedling dry weight in mg [57]. Two-way-ANOVA was used to analyse the statistical significance (*p* < 0.05) for germination and seedling data using Genstat (19th Edition, VSN International). 

### 4.4. Small RNA Sequencing

Total RNA was extracted from young seedlings using Tri reagent (Sigma-Aldrich, North Ryde, Australia) and treated with TURBO DNase (ThermoFisher Scientific, Scoresby, Australia) following the manufacturer’s instructions. The concentration and quality of the RNA samples were measured on a NanoDrop Lite spectrophotometer (ThermoFisher Scientific, Scoresby, Australia). RNA integrity was assessed by agarose gel electrophoresis and a Bioanalyzer (RIN > 9). Total RNA from eight biological replicates in each group was pooled for small RNA sequencing. Four small RNA libraries (TCG, TWH, SCG, SWH) were constructed using the TruSeq Small RNA Sample Prep Kit from the four groups AuCG, AuWH, L6CG, and L6WH, respectively. Single-end sequencing (50 bp) was performed on an Illumina HiSeq2500 by LC-Bio (Hangzhou, China). The sequencing datasets were deposited in NCBI GEO database (accession number GSE143454).

### 4.5. Analysis of Conserved and Novel miRNAs

Sequencing analysis was performed using the in-house ACGT101-miR program (LC Sciences, Houston, TX, USA). Raw sequencing reads were first processed by removing low-quality reads and adapter sequences to obtain clean sequences. The reads with a nucleotide (nt) length <18 nt or > 25 nt were removed. Common non-coding RNA families (rRNA, tRNA, snRNA and snoRNA), repeats and mRNA sequences were then discarded using RFam, Repbase and durum wheat NCBI mRNA entries as references. Unique sRNA sequences were then obtained for each library. Conserved mature miRNAs and 3p- or 5p-derived miRNA variants were identified by BLAST search against miRNA precursors in the miRBase (Release 22.1). MIR and miRNA sequences from common plant species in the miRBase were used as references (Appendix A). Single mismatch within the sequence and length variation at both 5p and 3p were allowed in the alignment. Sequences mapped to the mature miRNA in the hairpin were identified as conserved mature miRNAs. Sequences mapped to the opposite arm of mature miRNA in the hairpin were identified as 5p- or 3p-derived variants. All the mapped miRNAs were aligned to the durum wheat genome (NCBI UID 3439611, assembly Svevo.v1) to determine their genomic location.

The remaining unmatched sRNA sequences were used to identify novel durum miRNAs. Sequences were BLASTed to the durum wheat genome. To identify miRNA precursors, secondary hairpin structures containing matched sequences were predicted using RNAfold (http://rna.tbi.univie.ac.at/cgi-bin/RNAfold.cgi) from 120 nt of flanking genome sequences. The following criteria were used for the prediction of the secondary structure as previously described [35,58]: (1) the length of the hairpin (two stems and the terminal loop) is ≥50 nt; (2) the length of the hairpin loop region is ≤200 nt; (3) the cut-off of free energy is ≤−15 kCal/mol; (4) the stem region of the predicted hairpin is ≥16 bp in length; (5) for each bulge in the stem, the nucleotide length is ≤12 nt; (6) for each bulge in the mature region of the predicted hairpin, the nucleotide length is ≤4 nt; (7) the number of biased matches in each bulge in the mature region is ≤2; (8) the number of biased bulges in the mature region is ≤2; (9) the number of mismatches in the mature region is ≤4; (10) the mature region is ≥12 bp in length; and (11) the mature region in the stem is ≥80%. 

All miRNAs were categorised into five groups (G1 to G5). To analyse the miRNA expression profiles, data normalisation of the reads number was carried out as previously described [59]. Differentially expressed miRNAs (DEMs) were identified based on the normalised count. Chi-squared n × n test was used to detect DEMs when comparing TWH vs TCG vs SWH vs SCG. Chi-squared 2 × 2 test and Fisher’s test was used to detect DEMs when comparisons were made between two libraries (TWH vs TCG, SWH vs SCG, SCG vs TCG, SWH vs TWH). The cut-off for the *p* value was set as 0.01, 0.05 and 0.1 to generate Figure 4 and provide a total evaluation of the number of miRNAs that were up-/down-regulated. The cut-off for the *p* value was set as 0.05 for Appendix A to identify miRNAs with significant differential expression. DEMs with a |log2 (Fold change)| > 1 in each comparison were identified and highlighted if they were up/down-regulated in Appendix A.

### 4.6. Target Prediction and Functional GO Analysis

To predict protein-coding genes targeted by durum miRNAs, *in silico* analysis using the GSTAr was employed. Gene Ontology terms of the miRNAs targets were annotated according to their biological process, cellular component and molecular function. KEGG Pathways (http://www.genome.jp/kegg) of miRNA targets were also analysed. The statistical enrichment of GO terms and KEGG pathways was performed as previously described [35,60].

## Figures and Tables

**Figure 1 plants-09-00189-f001:**
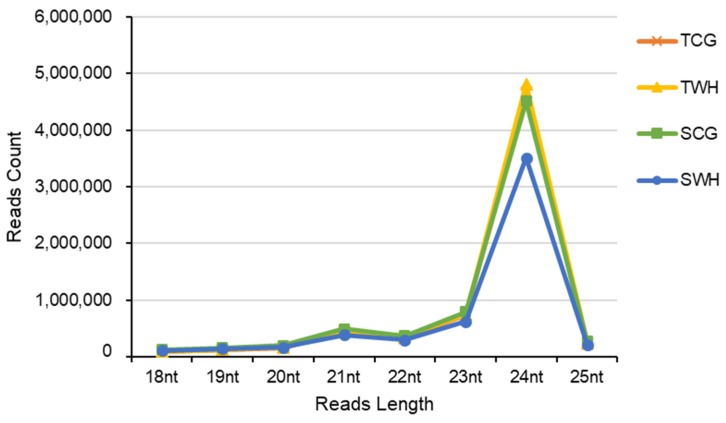
Small RNA size distribution of four durum wheat libraries. TCG, constructed from DBA Aurora CG seedlings. TWH, constructed from DBA Aurora WH seedlings. SCG, constructed from L6 CG seedlings. SWH, constructed from L6 WH seedlings.

**Figure 2 plants-09-00189-f002:**
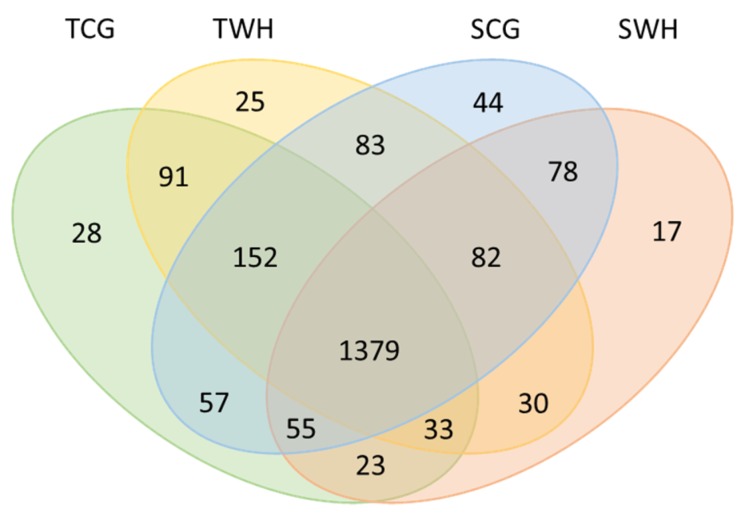
Venn diagram of all mature miRNAs in four libraries. TCG, constructed from DBA Aurora CG seedlings. TWH, constructed from DBA Aurora WH seedlings. SCG, constructed from L6 CG seedlings. SWH, constructed from L6 WH seedlings.

**Figure 3 plants-09-00189-f003:**
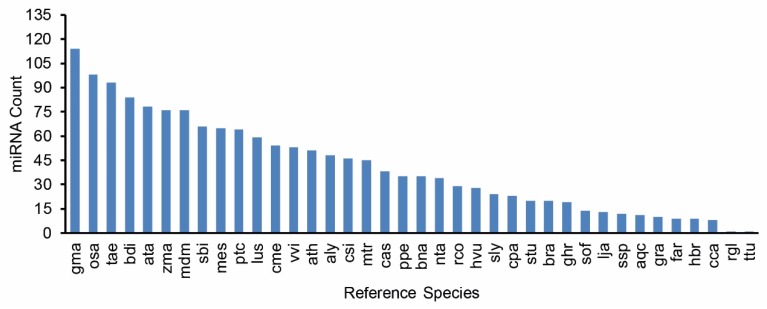
Ranking of reference plant species based on the number of miRNAs conserved. gma, *Glycine max*. osa, *Oryza sativa*. tae, *Triticum aestivum*. bdi, *Brachypodium distachyon*. ata, *Aegilops tauschii*. zma, *Zea mays*. mdm, *Malus domestica.* sbi, *Sorghum bicolor*. mes, *Manihot esculenta*. ptc, *Populus trichocarpa*. lus, *Linum usitatissimum*. cme, *Cucumis melo*. vvi, *Vitis vinifera*. ath, *Arabidopsis thaliana*. aly, *Arabidopsis lyrata*. csi, *Citrus sinensis*. mtr, *Medicago truncatula*. cas, *Camelina sativa*. ppe, *Prunus persica*. bna, *Brassica napus*. nta, *Nicotiana tabacum*. rco, *Ricinus communis*. hvu, *Hordeum vulgare*. sly, *Solanum lycopersicum*. cpa, *Carica papaya*. stu, *Solanum tuberosum*. bra, *Brassica rapa*. ghr, *Gossypium hirsutum*. sof, *Saccharum officinarum*. lja, *Lotus japonicas*. ssp, *Saccharum ssp.* aqc, *Aquilegia caerulea*. gra, *Gossypium raimondii*. far, *Festuca arundinacea*. hbr, *Hevea brasiliensis*. cca, *Cynara cardunculus*. rgl, *Rehmannia glutinosa*. Ttu, *Triticum turgidum*.

**Figure 4 plants-09-00189-f004:**
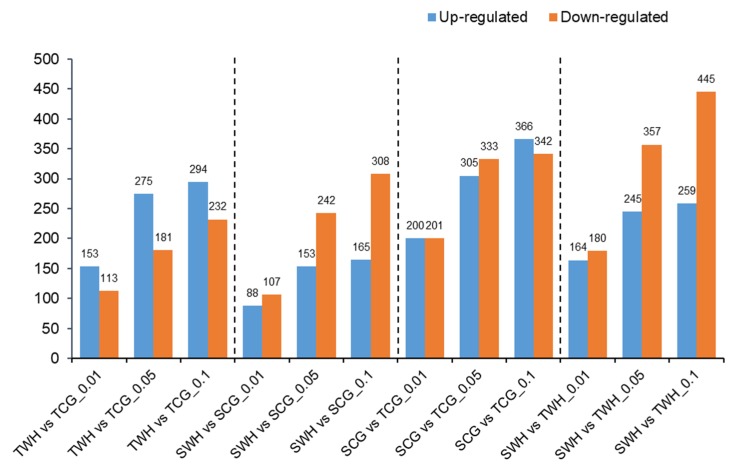
Number of up-regulated and down-regulated miRNAs in comparisons made between libraries at different *p* values (*p* < 0.01, *p* < 0.05 and *p* < 0.1). TCG, constructed from DBA Aurora CG seedlings. TWH, constructed from DBA Aurora WH seedlings. SCG, constructed from L6 CG seedlings. SWH, constructed from L6 WH seedlings.

**Figure 5 plants-09-00189-f005:**
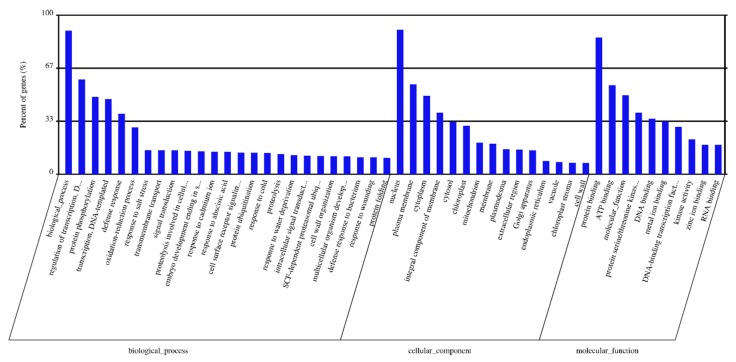
The most represented Gene Ontology terms among the predicted targets in three categories. GO terms were shown for each category based on the number of genes matched for each GO term (from higher to lower). Percent of genes (%) indicates the percentage of the gene number matched for each GO among the total number of genes within each category.

**Figure 6 plants-09-00189-f006:**
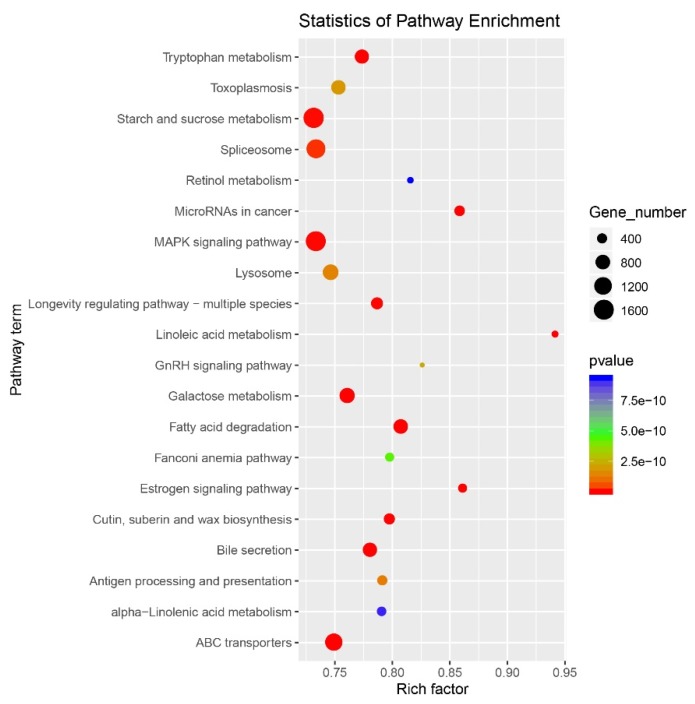
KEGG (Kyoto Encyclopedia of Genes and Genomes) pathway enrichment analysis of predicted target genes. *p*-value (red to blue colour) represents the significance of the matched gene ratio (rich factor). Dot size is proportionate to the number of target genes matched to the pathway term.

**Table 1 plants-09-00189-t001:** Germination potential (Gp, %), germination rate (Gr, %), germination index (Gi), coleoptile length, and young root length of the four seed groups. AuCG, seeds from DBA Aurora control group. AuWH, seeds from DBA Aurora stress group. L6CG, seeds from L6 control group. L6WH, seeds from L6 stress group.

Seed Source	Gp, %	Gr, %	Gi	Coleoptile Length (mm)	Young Root Length (mm)
AuCG	32.2% ± 1.1%	98.9% ± 1.1%	35.1 ± 0.8	31.88 ± 0.52	48.56 ± 1.06
AuWH	26.7% ± 1.9%	97.8% ± 1.1%	32.2 ± 0.9	31.06 ± 0.67	50.44 ± 1.19
L6CG	17.8% ± 1.1%	93.3% ± 1.9%	27.6 ± 0.2	30.38 ± 0.75	38.88 ± 1.28
L6WH	14.4% ± 1.1%	92.2% ± 1.1%	24.4 ± 0.6	30.50 ± 0.75	33.75 ± 1.23
F pr. Genotype	<0.001	0.004	<0.001	0.134	<0.001
F pr. Parent treatment	0.011	0.438	0.002	0.614	0.178
F pr. Genotype × Parent treatment	0.438	1.000	0.842	0.493	0.005
l.s.d Genotype	3.1%	3.1%	1.6	n.a	2.39
l.s.d Parent treatment	3.1%	n.a	1.6	n.a	n.a
l.s.d Genotype × Parent treatment	n.a	n.a	n.a	n.a	3.38

**Table 2 plants-09-00189-t002:** Root traits, shoot traits and seedling vigour indices of the four seed groups. AuCG, seeds from DBA Aurora control group. AuWH, seeds from DBA Aurora stress group. L6CG, seeds from L6 control group. L6WH, seeds from L6 stress group.

Seed Source	Root Fresh Weight (g)	Root Dry Weight (g)	Root Length (cm)	Shoot Fresh Weight (g)	Shoot Dry Weight (g)	Seedling Height (cm)	Seedling Vigour Index I	Seedling Vigour Index II
AuCG	0.718 ± 0.016	0.0807 ± 0.0016	25.86 ± 0.18	0.878 ± 0.025	0.1244 ± 0.0030	27.33 ± 0.26	1577.28 ± 30.48	202.80 ± 2.42
AuWH	0.736 ± 0.014	0.0812 ± 0.0022	28.46 ± 0.22	0.836 ± 0.022	0.1189 ± 0.0023	26.36 ± 0.23	1537.56 ± 28.01	195.63 ± 3.70
L6CG	0.710 ± 0.013	0.0721 ± 0.0017	25.10 ± 0.19	0.844 ± 0.020	0.1141 ± 0.0025	26.98 ± 0.24	1450.17 ± 21.74	173.78 ± 2.92
L6WH	0.658 ± 0.014	0.0645 ± 0.0014	26.56 ± 0.21	0.778 ± 0.016	0.1116 ± 0.0033	25.39 ± 0.31	1323.39 ± 16.81	162.32 ± 3.79
F pr. Genotype	0.005	<0.001	<0.001	0.034	0.004	0.018	<0.001	<0.001
F pr. Parent treatment	0.242	0.055	<0.001	0.015	0.160	<0.001	0.002	0.008
F pr. Genotype × Parent treatment	0.018	0.030	0.008	0.551	0.608	0.245	0.091	0.516
l.s.d Genotype	0.029	0.0036	0.41	0.043	0.0057	0.54	50.90	6.67
l.s.d Parent treatment	n.a	n.a	0.41	0.043	n.a	0.54	50.90	6.67
l.s.d Genotype × Parent treatment	0.041	0.0051	0.58	n.a	n.a	0.76	n.a	n.a

**Table 3 plants-09-00189-t003:** Overview of sequencing read counts in four sRNA libraries. TCG, constructed from DBA Aurora CG seedlings. TWH, constructed from DBA Aurora WH seedlings. SCG, constructed from L6 CG seedlings. SWH, constructed from L6 WH seedlings. Clean reads may not be equal to raw reads minus all the filter type reads, as there could be overlapped sequences between mRNA, Rfam and Repeats filters.

Read Type	TCG		TWH		SCG		SWH	
	Total Reads	Unique Reads	Total Reads	Unique Reads	Total Reads	Unique Reads	Total Reads	Unique Reads
Raw reads	20,042,833	8,293,425	20,900,186	8,593,357	24,262,343	8,865,724	18,786,468	7,221,909
3ADT & Length filter	6,000,364	1,418,893	5,651,835	1,475,235	7,669,010	1,832,961	6,589,933	1,676,502
Junk reads	92,538	68,473	101,066	73,192	111,339	79,088	76,371	56,322
Rfam - rRNA	215,716	5440	228,715	5378	308,312	6809	266,089	7444
Rfam - tRNA	230,895	2119	215,340	2207	248,091	2853	310,937	3964
Rfam - snoRNA	9600	393	9813	396	10,886	363	8009	304
Rfam - snRNA	2364	181	2581	194	3250	264	3580	215
Other Rfam RNA	13,680	733	14,276	706	19,445	886	17,165	893
mRNA	878,039	23,905	915,468	25,170	1,090,076	26,584	798,629	20,880
Repeats	6104	186	6073	193	9312	265	7704	243
Clean reads	12,631,656	6,773,722	13,795,248	7,011,288	14,830,872	6,916,493	10,781,672	5,456,127

**Table 4 plants-09-00189-t004:** Categorisation of all MIR-miRNA entries identified in this study. G1-G4 represent conserved miRNAs while G5 represents novel miRNAs. Unique miRNAs represent distinct mature miRNA products. G1, pre-miRNAs can be mapped to the durum genome and ESTs (expressed sequence tags). G2, sequencing reads mapped to the durum genome and ESTs with their extended genome sequences forming secondary hairpins. G3, sequencing reads mapped to the durum genome and ESTs, but their extended genome sequences do not form secondary hairpins. G4, neither aligned reference pre-miRNAs nor sequencing reads can be further mapped to the genome. G5, novel miRNAs where reads cannot be mapped to the miRBase but can be mapped to the genome, and secondary hairpins can be formed from extended genome locations. TCG, constructed from DBA Aurora CG seedlings. TWH, constructed from DBA Aurora WH seedlings. SCG, constructed from L6 CG seedlings. SWH, constructed from L6 WH seedlings.

Group	Total		TCG		TWH		SCG		SWH	
	Pre-miRNA	Unique miRNA	Pre-miRNA	Unique miRNA	Pre-miRNA	Unique miRNA	Pre-miRNA	Unique miRNA	Pre-miRNA	Unique miRNA
G1	161	270	158	252	158	256	158	259	158	257
G2	446	483	368	392	372	405	382	410	300	322
G3	838	890	663	694	701	736	722	763	611	644
G4	48	49	34	34	31	33	42	43	34	34
G5	437	485	426	446	429	445	429	455	421	440

**Table 5 plants-09-00189-t005:** Highly expressed differentially expressed miRNAs (DEMs) with a |log2 (fold change)| > 1 in each comparison made between two libraries (TWH vs TCG, SWH vs SCG, SCG vs TCG, SWH vs TWH). TCG, constructed from DBA Aurora CG seedlings. TWH, constructed from DBA Aurora WH seedlings. SCG, constructed from L6 CG seedlings. SWH, constructed from L6 WH seedlings.

**miR_name**	**miR_seq**	**up/down**	**log2(FC)**	**TCG**	**TWH**
osa-miR168a-3p_L-3	CCCGCCTTGCACCAAGTGAAT	up	2.04	3571	14,660
ata-miR169a-3p	TGGGCAAGTCACCCTGGCTACC	up	1.11	533	1154
**miR_name**	**miR_seq**	**up/down**	**log2(FC)**	**SCG**	**SWH**
tae-miR9661-5p_1ss14GA	TGAAGTAGAGCAGAGACCTCA	down	−1.06	3513	1689
tae-miR9774_L+2	AACAAGATATTGGGTATTTCTGTC	up	1.27	1589	3841
**miR_name**	**miR_seq**	**up/down**	**log2(FC)**	**TCG**	**SCG**
ata-miR167b-5p	TGAAGCTGCCAGCATGATCTGA	down	−1.21	20,089	8687
tae-MIR7757-p3_1ss19AG	TGGATAGTTTGAGGTTTTGTTT	down	−1.04	6524	3162
ata-miR167b-3p	AGGTCATGCTGGAGTTTCATC	down	−1.11	5746	2653
ttu-miR160	TGCCTGGCTCCCTGTATGCCA	down	−1.04	4739	2304
mtr-miR398a-5p_1ss7AC	GGAGTGCCACTGAGAACACAAG	down	−1.36	2354	917
csi-miR166c-3p_R+1	TCGGACCAGGCTTCATTCCCA	down	−1.52	1062	370
tae-miR9654b-3p_1ss19GA	TTCCGAAAGGCTTGAAGCAAAT	down	−1.10	1542	721
ata-miR396e-3p	GTTCAATAAAGCTGTGGGAAA	down	−1.10	737	345
sbi-miR396c_R+1	TTCCACAGCTTTCTTGAACTTT	down	−1.07	758	361
PC-5p-1206_1154	TTTGCGCAGAAGGGAGAAATC	up	inf	0	2228
tae-MIR5048-p3_1ss18TG	AATATATTTGCAGGTTTGAGG	up	1.08	4677	9898
ata-miR169f-3p_R-1	GGCAAGTCCGTCCTTGGCTAC	up	1.22	4609	10,772
osa-miR168a-3p_L-3	CCCGCCTTGCACCAAGTGAAT	up	1.26	3571	8570
ata-MIR169d-p3	TTGTCCTTGGCTACACCTAGT	up	1.50	1742	4936
tae-miR9661-5p_1ss14GA	TGAAGTAGAGCAGAGACCTCA	up	3.49	312	3513
ata-MIR396d-p3	AAGCCCATGGAAACCATGCCC	up	4.66	29	735
ata-miR169i-5p_1ss15TC	TAGCCAAGGATGACCTGCCTG	up	3.39	53	553
ata-miR169a-3p	TGGGCAAGTCACCCTGGCTACC	up	1.49	533	1494
PC-5p-726_2034	TACGGCAAAGCCGTCGGCATA	up	2.03	194	793
ata-miR169h-3p_L-1	CAAGTTGTTCTTGGCTAGC	up	1.25	484	1154
tae-MIR164-p3	CATGTGCCTTTCTTCTCCACC	up	1.02	419	848
**miR_name**	**miR_seq**	**up/down**	**log2(FC)**	**TWH**	**SWH**
mtr-miR398a-5p_1ss7AC	GGAGTGCCACTGAGAACACAAG	down	−1.69	1516	471
tae-miR9659-3p	TCCAATGGTTGTTCACGGCATC	down	−1.17	910	404
osa-miR160f-5p_L-2R+2	CCTGGCTCCCTGAATGCCATC	down	−1.14	817	370
sbi-miR396c_R+1	TTCCACAGCTTTCTTGAACTTT	down	−1.12	609	280
PC-5p-1206_1154	TTTGCGCAGAAGGGAGAAATC	up	inf	0	2473
ata-miR169f-3p_R-1	GGCAAGTCCGTCCTTGGCTAC	up	1.14	4103	9018
tae-miR9774_L+2	AACAAGATATTGGGTATTTCTGTC	up	1.57	1296	3841
tae-miR9661-5p_1ss14GA	TGAAGTAGAGCAGAGACCTCA	up	2.55	288	1689
ata-MIR396d-p3	AAGCCCATGGAAACCATGCCC	up	4.97	23	710
ata-miR169i-5p_1ss15TC	TAGCCAAGGATGACCTGCCTG	up	2.52	82	469

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
