# Peer review of "Transgenerational Effects of Water-Deficit and Heat Stress on Germination and Seedling Vigour—New Insights from Durum Wheat microRNAs"

_plants, 2020, doi:10.3390/plants9020189_

Round 1

Reviewer 1 Report

The work provides new information in the stress epigenetics of durum wheat. The authors sequenced the microRNAome of stress tolerant and sensitive durum wheat genotypes and identified several differentially expressed candidate miRNAs. The bioinformatic analysis and the discussion on the biological function of the possible targets is sound and thereby a working model has been built.

Can the authors provide genetic evidence,  e.g. genome sequence data (like promoter sequence) of the tolerant and sensitive genotypes which can explain explain the cause of the differential expression of the major microRNAs,  as described in the discussion? 

Author Response

Dear Editor,

Please find below our comments to the Reviewers points. We look forward to hearing from you on the outcome of this paper.

Best regards,

Haipei

Reviewer 1.

We thank Reviewer 1 for acknowledging the scientific soundness of our research and the constructive advice. We have revised the Discussion based on Reviewer 1’s comments. Indeed, genomic information of the upstream regulatory components of durum miRNAs (such as the promoter region) can provide further information on the transcriptional regulation of miRNA expression. However, such information cannot be extracted from the sRNA next-generation sequencing approach used in this study. The full genome information of the stress-tolerant and -sensitive Australian genotypes are not available. Currently the durum genome sequence was extracted from the Italian cultivar Svevo. Still, the Reviewer’s valuable comments point direction to important future research where focus can be placed on genomic analysis of the promoter regions surrounding major stress-responsive miRNA transcripts and the organisation of miRNA clusters using experimental approaches such as RACE methods. Our current research has paved the way for such research, having provided in-depth details of the MIR gene (genome ID, length, pre-miRNA sequence, secondary structure etc.) in Supplementary Table S1. Please see revision made in Lines 414-422.

Reviewer 2 Report

The manuscript presented the results from the roles of miRNAs on the regulation of the germination of the durum wheat seeds under water-deficient and heat conditions. The miRNA data were well analysed and presented. The manuscript will be valuable for durum wheat researchers and to be published.

Some minor changes are required as below.

L6CG was used for describing L6 stress group in a number of lines, including line 95, 430, and 493, and the tiles of Table 1 to 4. L493 did not clearly describe how to normalise the number of reads for each library. Such information cannot be found in reference 58 as cited in the manuscript.

Author Response

Reviewer 2.

We thank Reviewer 2 for the helpful suggestions. We apologise for these oversights. The misuse of L6CG has been corrected to L6WH (line 95, 124, 128, 438). We have added the correct reference that describes the details of data normalisation in Line 501. The details are cited below from the new reference Li et al. 2016.

Data normalization followed the procedures as described in a previous study [69] with minor modification. (1) Find a common set of sequences among all samples; (2) Construct a reference data set. Each data in the reference set is the copy number median value of a corresponding common sequence of all samples; (3) Perform 2-based logarithm transformation on copy numbers (Log2 (copy#)) of all samples and reference data set; (4) Calculate the Log2 (copy#) difference (ΔLog2 (copy#)) between individual sample and the reference data set; (5) Form a subset of sequences by selecting |ΔLog2 (copy#)| < 2, which means less than 4 (22) fold change from the reference set; (6) Perform linear regressions between individual samples and the reference set on the subset sequences to derive linear equations y = aix + bi, where ai and bi are the slop and interception, respectively, of the derived line, x is Log2 (copy#) of the reference set, and y is the expected Log2 (copy#) of sample i on a corresponding sequence; (7) Calculate the mid value xmid = (max(x) - min(x))/2 of the reference set. Calculate the corresponding expected Log2 (copy#) of sample i, yi,mid = aixmid + bi. Let yr,mid = xmid, let Δyi = yr,mid - yi,mid, which is the logarithmic correction factor of sample i. We then derive the arithmetic correction factor fi = 2Δyi sample I; (8) Correct copy numbers of individual samples by multiplying corresponding arithmetic correction factor to original copy numbers.

Reviewer 3 Report

Review of the manuscript entitled "Transgenerational effects of water-deficit and heat stress on germination and seedling vigour – new  insights from durum wheat microRNAs" from Liu et al. submitted for publication in Plants (MDPI).

Abiotic stresses are the major constraints to wheat cultivation worldwide. Such stresses could also have transgenerational negative effect on the next progeny. In this work the authors investigated the influence of the maternal environment (water deficit and heat stress) on the offspring early growth in two Australian durum genotypes (one stress-tolerant and one stress-sensitive). They showed that stress on the previous generation negatively affects seed germination and seedling vigour in both genotypes, even with a lesser extent in the stress-tolerant one. More importantly, the authors provided the first report of transgenerational effects of stress in durum wheat at epigenetics level (miRNA expression) exploiting the recently available durum wheat genome assembly (Svevo.v1). They successfully used bioinformatics approaches to analyse durum miRNAome and to identify candidate miRNA target genes and pathways that could lead to better understanding of the epigenetic mechanisms underlying the transgenerational effect of stress in this important cereal crop.

This is very interesting and profound work. The manuscript is well written and describe all the analysis clearly. The used methods are appropriate. The data and results are solid and discussed properly.

Minor comments:

1.       Line 16: “durum genotype” should be “durum genotypes”.

2.       Line 72: “NGS (next generation sequencing)” better to say “Next Generation Sequencing (NGS)”.

3.       Lines 84-85: “the new durum genome assembly (Svevo.v1, available 2019) for epigenetic studies [33], providing a more accurate, in-depth analysis on the whole genome-scale” better to say “the new durum genome assembly (Svevo.v1, available 2019 [33]) for epigenetic studies, providing a more accurate in-depth analysis on the whole genome-scale”.

4.       Lines 95-96: “L6CG (seeds from L6 water-deficit and heat stress group)” should be “L6WH (seeds from L6 water-deficit and heat stress group)”.

5.       Line 101: “Parent treatment only had significant impact (P < 0.05) on Gp and Gi”. Table 1 shows that parent treatment has significant impact only on Gi when P < 0.05. Please check that.

6.       Comparing the values of “young roots length” and “root length” in Table 1 and Table 2, respectively, shows that “young roots length” is always longer than “root length”. Why? As I understood from “Materials and Methods” I think that “root length” should be longer. Please check that.

7.       Line 111: “great” better to say “greater”.

8.       Line 124-125: “L6CG, seeds from L6 stress group” should be “L6WH, seeds from L6 stress group”.

9.       Line 127-128: “L6CG, seeds from L6 stress group” should be “L6WH, seeds from L6 stress group”.

10.   In Table 1 and Table 2, please replace “<.001” by “<0.001”.

11.   Line 132: “A total of 83.99 million” please write the exact number (83,991,830) or “A total of ca. 83.99 million”.

12.   Line 133: “Approximately 52.04 million clean sRNA reads”. According to the values shown in Table 3, this number should be 51.85. I suppose that you calculate the “Clean reads” values in Table 3 as: “Clean reads” = “Raw reads” – (“3ADT & Length filter” + “Junk reads” + “Rfam – rRNA” + “Rfam – tRNA” + “Rfam – snoRNA” + “Rfam – snRNA” + “Other Rfam RNA” + “mRNA” + “Repeats”). If so, then there are calculation errors or maybe some missing values that are not shown in the table. Please check that.

13.   Line 133: “18-25nt” should be “18-25 nt”.

14.   Line 150: “unique miRNAs” could you please provide a definition for this term, which will help in understanding Table 4.

15.   Line 153: in the caption of Table 4: “all MIR-miRNA entries” I think this refers to the 2202 MIR-miRNA entries (detailed in the Supplementary Table S1). It was difficult for me to find a connection between the numbers in Table 4 and the data provided in Table S1. Could you please add more details in Table S1 (or in a separate supplementary table) that show which of the MIR-miRNA entries are “Unique miRNA” and “Pre-miRNA” for each of the four libraries?

16.    Line 169: “2176” this number correspond with the sum of the numbers shown in the Venn diagram (Figure 2), but not with the numbers in Table 4 (where it should be 2177). Please check which one is wrong and correct it.

17.    Line 181: “mdm,Malus domestica” should be “mdm, Malus domestica”.

18.    Line 195: “TCG and TWH” should be “TCG and/or TWH”. If you want to keep it as “(TCG and TWH)” then the total number of miRNAs that are exclusively expressed in DBA Aurora should be 91 instead of 144.

19.    Line 196: “SCG and SWH” should be “SCG and/or SWH”.

20.    Lines 197-198: “TCG and SCG” should be “TCG and/or SCG”.

21.    Lines 198: “TWH and SWH” should be “TWH and/or SWH”.

22.    In Figure 4, I think there is no point of showing the number of DEMs at P < 0.1 (the statistical difference at this P value is not significant).

23.    In Figure 4, I think there is a labelling mistake in the X axis of the fourth comparison group (SWH vs TWH_0.1; SWH vs TWH_0.01; SWH vs TWH_0.05). Please make sure that the labelling is right.   

24.    Line 223: “A total of 311,964 transcripts”. Please mention also the number of potential target genes (gene IDs).

25.    In Figure 5: It is not clear the criteria used to select the GO terms shown in this figure. Are they the GO terms with the smallest enrichment P value or the biggest number of the genes? What does “Percent of genes” mean? Please provide this information in the caption of this figure.  

26.    In Supplementary Table 8: “TB gene number: total number of significant targets that match to GO terms” should be “TB gene number: total number of targets that match to GO terms”.

27.    Lines 242-246: “The top-ranked enriched pathways are shown in Figure 6. In particular, a total of 2248 targets were matched to starch and sucrose metabolism (ko00500), 2171 targets were matched to the MAPK signalling pathway (ko04010), 1191 target were matched to the spliceosome (ko03040), and 1720 targets were matched to ABC transporters (Supplementary Table S9)”. Why “Protein processing in endoplasmic reticulum (ko04141)” that contains the biggest number of targets (2406 targets) is not mentioned?

28.    In Figure 6: Again, the pathway “Protein processing in endoplasmic reticulum (ko04141)” is missing.. Why?  

29.    Line 427: “exposing the plants to 37 °C/27 °C for 24h at 5, 15, 25, 35, and 45 days”. Here, there are 5 different treatment. Please specify from which treatment the seeds in AuWH and L6WH were taken for the germination test and the small RNA sequencing.      

30.    Line 430: “L6CG, seeds from L6 stress group” should be “L6WH, seeds from L6 stress group”.

31.    Lines 431: “Germination test”: You mentioned previously that under water-deficit stress and heat stress DBA Aurora produced bigger grains that were full-looking, while L6 had significantly smaller seeds with reduced weight. Considering that seed size/weight influences significantly seed vigour, it’s important to conduct the germination experiment on seeds with similar size/weight from both genotypes to show that the observed differences are purely due to transgenerational effect of stress, but not due to other factors (as grain weight). Did you considered this point in your experiment? If so, please mention that in this paragraph.

32.    Line 461: “the four groups: AuCG, AuWH, L6CG, and L6WH” should be ““the four groups AuCG, AuWH, L6CG and L6WH, respectevely”.

33.    Lines 480-491: Is there any reference for the used parameters? I think that “mature region is ≥12 bp in length” is too small.

34.    Line 492: “(G1-G5)” better to say “(G1 to G5)” or “(G1, G2, G3, G4 and G5)”.

35.    Lines 497-498: “The significance threshold was set to be 0.01, 0.05 and 0.1”. Does this threshold mean the significance level at which you reject H0? I think that 0.1 (less than 1 in 10 chance of being wrong) is quite big.
